# Institutional Ambiguity and Ontological Politics in Integrating Sustainability into Finnish Dietary Guidelines

**Minna Santaoja** [1,*] and **Mikko Jauho** [2]

1   Finland Futures Research Centre, University of Turku, 20014 Turku, Finland
2   Centre for Consumer Society Research, University of Helsinki, 00014 Helsinki, Finland;
    mikko.jauho@helsinki.fi
*   Correspondence: minna.santaoja@utu.fi

**Abstract:** Dietary guidelines are a key policy instrument in guiding the way people eat in many countries. Traditionally, the guidelines focus on the public health aspects of diets. During the last decade, sustainability has increasingly been incorporated into dietary guidelines, emphasizing that sustainable diet benefits both health and the environment. This article analyses the integration of sustainability into dietary guidelines in Finland. The analysis is situated within the ontological turn in social theory, understanding food as ontologically multiple. We employ Annemarie Mol's concept of ontonorms in analyzing the Finnish dietary guidelines. Currently, in Finland, there seems to be a situation of institutional ambiguity regarding where and by whom sustainable food policy is being made and what does it constitute. We claim that the ontological multiplicity of food is partly constituted by, and at the same time constitutive of, the institutional ambiguity, and as a result, the guidelines do not yet provide clear guidance for sustainable food practices. As the guidelines fail to coordinate the multiplicity, they increase the normative burden on consumers to make responsible choices. In the latest Finnish guidelines targeted for children, however, steps are taken towards a more inclusive, caring understanding of sustainable dietary guidance.

**Keywords:** dietary guidelines; sustainable diet; ontological politics; ontonorms; food policy; institutional ambiguity; plant-based diet; Finland

## 1. Introduction

National and other official dietary guidelines are a powerful policy instrument in guiding the way the population eats. Traditionally, the guidelines focus on health, defining the composition of a healthy diet on the population level. Such guidelines have an established institutional status, for instance, in the Nordic countries, as key policies addressing food-related health. During the last decade, the so-called ecological public health perspective [1] has increasingly been incorporated into dietary guidelines, emphasizing that sustainable diet benefits both health and the environment [2]. For instance, the Dutch dietary guidelines [3] include a brief discussion on ecological aspects of diet, the Swedish guidelines [4] have incorporated a sustainability perspective throughout, and the UK guidelines (the Eatwell Guide) have been evaluated for their environmental impacts [5]. The most ambitious attempt to specify a universal healthy reference diet within planetary boundaries was recently provided by the EAT-Lancet Commission [6], which developed a framework for healthy diets from sustainable food production. In these proposals and related discussions, the guidelines are presented as a tool for nutrition education, integrating health and environmental sustainability [6–8]. However, as the shift towards sustainable dietary guidelines has important repercussions for food production, it is also contested [9,10].

In this article, we analyze the integration of environmental sustainability into nutrition guidelines in the Finnish context, where the latest edition of the guidelines [11] discusses the sustainability of food choices. In Finland, the guidelines are authored by the National Nutrition Council, an official expert body under the Ministry of Agriculture and Forestry. Policy documents such as nutrition guidelines are constitutive of interventions, procurement, and product development. The Finnish guidelines are non-binding, yet are important in setting benchmarks for nutritional targets for various population groups and for favoring certain foods over others. In Finland, the guidelines do not primarily address citizens directly but are rather aimed at professionals providing dietary advice, making purchases and planning menus in public mass catering, and education. The guidelines can be characterized as a hybrid of a nutrient—and meal—oriented approach [12]. They become translated to the individual level, as the principles stipulated in the guidelines are used in individual dietary counselling. The nutrition guidelines gradually become household vocabulary, affecting the self-perception of the citizens concerning food and eating. Besides the general guidelines, the National Nutrition Council has issued guidelines to various population groups with specific nutrition needs, such as the elderly and families with children [13], and for specific contexts, such as school meals [14].

The Finnish dietary guidelines are set in a broader food policy context. The national guidelines in Finland and other Nordic countries build on the Nordic Nutrition Recommendations that have been produced in Nordic collaboration every eight years since 1980. The primary aim of the Nordic recommendations is to present the scientific background of the guidelines and to function as a basis for the national recommendations of the various countries [15]. The next Nordic recommendations are due in 2022 [16]. Alongside the nutrition recommendations, a new integrated food policy is taking shape. In 2010, a national food strategy was drafted for Finland, and in 2016, it was replaced by the Government Report on food policy, "Food 2030" [17], which takes a systemic approach and also explores the environmental impacts of food production and consumption. Also, in 2016, a Food Policy Committee was established by the Ministry of Agriculture and Forestry to develop and coordinate the implementation of food-related policies in central government. The National Nutrition Council was placed under the Food Policy Committee as one of three sub-committees. However, the Food Policy Committee was appointed only for a fixed term of three years, until August 2019. The long history of the National Nutrition Council has established it as an authority in food policy, and the 2014 dietary guidelines were still introduced as forming the basis of Finnish food policy [11] (p. 5). Thus, there seems to currently be a situation of institutional ambiguity [18] regarding where and by whom food policy is being made and what does it constitute. Institutional ambiguity refers to the boundary area between institutional settings, where new rules are needed to bring the various institutions and policies together [19]. As such, while institutional ambiguity may negatively affect the effectiveness of policymaking and implementation, it may also be a fruitful situation where dialogue, debate, and new practices may emerge.

In this paper, we analyze how this institutional ambiguity is reflected in the inclusion of sustainability in the Finnish dietary guidelines. We carry out an interpretive policy analysis [20], and by focusing on the actors, events, and views around sustainability in the recommendations, we assess what aspects of sustainability are emphasized, and what the enactment of sustainability implies for the proposed food choices. We place our analysis within the "ontological turn" in social theory, which claims that discourses and language alone are not enough to understand social reality [21]. Besides a policy object, food is part of everyday material reality in various ways. The critical role of everyday practices in the transition towards sustainability has been highlighted by numerous studies (e.g., [22,23]). Because the practices are plural, they are ontologically enacting different realities, in which food is configured in ontologically plural ways. In our analysis, we draw on Annemarie Mol's [24,25] notion of ontonorms, which refer to the entanglement of ontological and normative commitments in various dietary techniques: what kinds of food and bodies the techniques enact, and what kind of normative commitments ensue from these enactments. While Mol's focus was on food and diet, the concept has recently been put to use also in other domains [26,27], and here we

explore its use in policy analysis. In order to unpack the institutional ambiguities of sustainable food policy, it is necessary to reflect not only on the path dependencies and the multiplicity of perspectives involved [28], but also to understand the ontonorms enacted in policies. We claim that the ontological multiplicity of food in the dietary guidelines is partly constituted by, and at the same time constitutive of, the institutional ambiguity regarding sustainable food policy in Finland. As a result, the policy does not provide clear guidance on sustainable food practices.

Food is one of the key sustainability issues globally, and many countries are facing the task of including sustainability considerations in nutrition and dietary recommendations. Thus, while our analysis focuses on a small Northern European country, we believe the analysis and the results to be of wider interest for analyzing and developing sustainable food policies.

In the next section, we discuss the theoretical approach of ontological politics more in detail and then characterize our key concept, ontonorms. The following section presents our research materials and methodological approach. In the first empirical section, using stakeholder interviews, we outline the process of establishing dietary guidelines and the introduction of sustainability into it. The second empirical section identifies the key ontonorms in dietary guidelines upon which the new sustainability perspective builds. The following two empirical sections look more closely at how environmental sustainability is framed in the Finnish 2014 dietary guidelines, and what kind of ontonormativity these framings enact. While the sustainability perspective is an important and promising opening, it sits somewhat uneasily alongside the traditional health emphasis and does not form a coherent whole. The final empirical section discusses emerging alternative ways of enacting sustainable eating, drawing from the more recent, group-specific dietary guidelines. We conclude by pointing to the interrelationship of institutional ambiguity and ontological multiplicity. Thus, while our focus is on the 2014 Finnish guidelines where the sustainable choices on the plate were introduced, we look also backwards to older guidelines and forwards to the most recent, group-specific guidelines for reference.

## 2. The Ontological Politics and Ontonorms of Dietary Guidance

The concepts of ontological politics and ontonorms pertain to the so-called ontological turn in social research. It represents a reaction to the preceding 'linguistic turn,' which is criticized for relying excessively on language to understand social relations [20]. The result is a renewed interest in materialisms of various guises. According to Annemarie Mol [29], a key proponent, ontological politics denotes a distancing from both perspectivism and constructivism. It is not a question of different actors' necessarily partial standpoints on the same object, nor the historical carving out and present support for a singular but contingent truth. Mol subscribes to constructivism but brings it further, claiming that instead of a singular world being constructed, there are multiple realities enacted simultaneously. The world is, thus, ontologically multiple [23], i.e., different practices continuously enact different realities, which need to be coordinated to present coherence. As such, the ontological turn in social theory is related to another refocusing, namely the "turn to practice." Now, if we grant that reality is multiple and dependent on the practices we endorse, it makes a difference which practices we choose to cultivate—the choice is political [23,24].

The various turns in social theory also have their critics. Regarding the ontological turn, criticism has been targeted on the treatment of practices as self-evident givens [25] and the "onto-theological" rigidity of some ontological approaches [30]. This kind of pre-defined ontological stance may lead to theoretical path-dependency, instead of valorizing different ontologies, which is the explicit aim. However, we consider the approach by Mol and colleagues on the multiple ontologies of food [24,31] to be flexible enough to steer clear from these dangers, as Mol refused to develop any "theory of ontonorms" but argued for theoretical fluidity and specificity.

In her analysis of food practices, Mol introduced the concept of ontonorms. Within the wider framework of ontological multiplicity and politics, the concept foregrounds the normative aspects of different enactments of reality. Mol [24] followed the practices of dietary professionals attending to people struggling with being overweight, and recognized three dietary techniques in dietary advice:

counting calories, making lists of more or less healthy foods, and models of balanced meal composition, such as the plate model. Each of the techniques stems from a different scientific tradition, enact food and the body to be ontologically different, and impose different norms as to how to eat. In counting calories, food is perceived as fuel and eating is approached with a biophysical model, where the key is to balance the intake and output of the energy food provides. Secondly, distinctions between good and bad foods are based on epidemiological research on the health effects of various food items, which dictates which ones should be promoted and which ones avoided in the diet. Finally, the models of meal composition, such as the Dutch disc of five or the Finnish plate model, demonstrate a biochemical approach, which focuses on sufficient and balanced intake of various nutrients. The plate model, for instance, shows how the requirements may be met on a single meal.

According to Mol, the techniques share a common normative message "Mind your plate." The mainstream dietary advice is based on the presupposition that while people often know how to eat "right," they are unable to resist their impulses, which leads to public health problems. Healthiness and desirability of food are staged as being in tension. A person who wants to eat wholesomely and lose weight needs to overrule the desires of the craving body. Thus, the dieting advice aims at providing cognitive tools for taking control of the body. As a contrasting figure, Mol [24] presented a more marginalized alternative to the "Mind your plate" advice, the message to "Enjoy your food," discussed in more detail by Vogel and Mol [31]. The message takes a more emphatic approach to food and eating, stressing the cultivation of culinary skills and personal tastes. Enjoyment, thus, also requires specific techniques and sensitivities. One needs to learn to attend to food and the body and to master new practices in shopping and selecting food, cooking, and looking after one's surroundings, close relatives, and friends as a localized and situational practice. While the "Mind your plate" advice adopts the logic of choice, "Enjoy your food" follows the logic of care [32]: instead of thinking "Am I being a good food citizen?" we are encouraged to think "Is this good for me?". Gearing dietary advice towards creating capabilities in care for the self and others might, in the long run, be more efficient than the approach stressing cognitive control and individual choice [24,31].

In this article, we move away from consultation rooms and face-to-face interaction between dieticians and their clients and address dietary guidance on another level. We extend the approach to policy analysis and investigate dietary guidelines as normative policy documents that define principles followed in the various contexts of food provision, including dietary counselling. Our interest is to see what kind of ontonorms the Finnish nutrition guidelines enact, and especially what kind or ontonormativity that the inclusion of sustainability enacts in the guidelines. More precisely, our research questions are: (1) how do the nutrition guidelines position citizen-consumers with regard to sustainable diets, and (2) what is the role of nutrition guidelines within broader sustainable food policy. Our theoretical contribution is to apply the perspective of ontological politics and ontonorms in policy analysis, directing attention from language and multidisciplinary epistemologies towards food as ontologically multiple, and the different ontological commitments that need to be negotiated for effective guidance on sustainable diets.

## 3. Materials and Methods

Our first research material consists of twelve semi-structured thematic interviews with current and former members of the Finnish Nutrition Council, the official body issuing the dietary guidelines. The interviews were carried out by the first author during summer 2017, and were recorded and transcribed verbatim by an external service provider. The interviews lasted from approximately an hour to 1.5 h, resulting in 128 pages of transcriptions. The interviews were held in Finnish, and quotes in the text have been translated by the authors. We interviewed people who have had a central role in the work of the Council and who participated in preparing the latest population-level guidelines from 2014, introducing sustainability. In identifying the interviewees, we used the snowballing method, asking the first identified key interviewees who could provide us with further insights regarding the guidelines, from different perspectives. The interviewees included health and social affairs authorities,

agriculture and forestry authorities, and researchers (primarily nutrition scientists), as well as food industry representatives. We agreed to maintain the anonymity of the interviewees, and for this reason, will refer only rather vaguely to their institutional or professional background.

Our second research material (see Table 1) consists of the general Finnish nutrition guidelines from 1987, 1998, 2005, and 2014. The documents from 2010 onwards are available on the web pages of the Finnish Food Authority [33]. The older documents were obtained as paper copies from the archives of the Food Authority by a project assistant. Several group—and context—specific guidelines complement the population-level food and nutrition guidelines. Our material includes the dietary recommendations for families with children from 2016 and the meal recommendations for schools from 2017. These are also available in English, unlike the general guidelines. Altogether, the document materials added up to approximately 490 pages for analysis.

**Table 1.** Nutrition and food policy documents analyzed, with primary material highlighted.

| Year | Title | Document Type |
|------|-------|---------------|
| 1987 | Valtion ravitsemusneuvottelukunnan mietintö. Suositukset kansanravitsemuksen kehittämiseksi. Osa I. Yleiset suositukset. Osa II. Yksityiskohtaiset suositukset ja perustelut. (Report of the National Nutrition Council. Recommendations for improving public nutrition. Part I. General recommendations. Part II. Detailed recommendations and arguments.) [34] | National nutrition recommendations |
| 1998 | Suomelaiset ravitsemussuositukset. Komiteamietintö 1998:7. (Finnish nutrition recommendations. Committee report.) [35] | National nutrition recommendations |
| 2005 | Suomalaiset ravitsemussuositukset—ravinto ja liikunta tasapainoon. Valtion ravitsemusneuvottelukunta. (Finnish nutrition recommendations—nutrition and physical exercise in balance. National Nutrition Council.) [36] | National nutrition recommendations |
| 2012 | Nordic nutrition recommendations 2012. Integrating nutrition and physical activity. Nordic Council of Ministers. [37] | Nordic nutrition recommendations |
| **2014** | **Terveyttä ruoasta! Suomalaiset ravitsemussuositukset 2014. Valtion ravitsemusneuvottelukunta. (Health from food! Finnish nutrition recommendations 2014. National Nutrition Council.) [11]** | **National nutrition recommendations** |
| 2016 | Ruoka 2030. Suomi-ruokaa meille ja maailmalle. Valtioneuvoston selonteko ruokapolitiikasta. (Food 2030. Finland feeds us and the world. Government report on food policy. Ministry of Agriculture and Forestry) [17] | Government report on Finnish food policy |
| 2017 | Eating and learning together—recommendations for school meals. National Nutrition Council, Finnish National Agency for Education, National Institute for Health and Welfare. [14] | Finnish nutrition recommendations for schools |
| 2019 | Eating together—food recommendations for families with children. National Institute for Health and Welfare in Finland. (First Finnish edition 2016) [13] | Finnish nutrition recommendations for families with children |

We utilized a qualitative content analysis to analyze the empirical materials [38]. Qualitative content analysis is a suitable approach when the focus of analysis is on the content of language, and the approach can be used in different ways depending on the aims and scope of the study. The interviews were analyzed by several rounds of theoretically informed close reading, highlighting sections addressing the research questions and organizing the contents thematically. The analysis of the document material took place similarly through several rounds of close reading of the documents, highlighting aspects related to a sustainable diet, and thematically organizing sections from the recommendation text. Our primary focus was first on the dietary guidelines from 2014. In our reading of the material, we focused on how and in what context sustainability is spoken of, how it is defined, how health and environmental sustainability are combined, and what kind of choices sustainable diets are seen to entail. We then went back to the older recommendations (1987, 1998, 2005), looking at whether and how sustainability themes were visible earlier already, and finally, we turned to the more recent recommendations for families (2016) and schools (2017) to see how the theme of eating sustainably was further elaborated in these group-specific guidelines.

Our reading of the various ontonorms and the corresponding dietary techniques in the guidelines, presented in Section 5, were developed in several iterative rounds of cross-reading between the authors. Our emphasis was especially on the ontonorm of sustainable eating that seems to be in the making in the Finnish dietary guidelines. The interviews shed light on the process of compiling the guidelines and introducing the sustainability perspective among varying stakeholder interests. The Council members differed in their understanding of sustainable eating and its role in the guidelines. The interviews thus shed light on the "ruling relations" [39] among them, which shaped the guidelines' take on a sustainable diet. The interviews allow us to take a broader interpretative policy analysis lens, to understand some of the choices made in the guidelines, and to place the guidelines into the field of food policy.

## 4. Making Dietary Guidelines: Institutional Ambiguity in Food Policy

The first comprehensive, population-wide dietary guidelines were released in Finland in 1987. First, the Finnish nutrition guidelines establish target values for the various vital nutrients necessary for optimal health and well-being, based on the best available scientific knowledge. Second, they translate these target values into food recommendations, as in the form of the food pyramid and plate model that stipulate how the various nutrients can be consumed. While the target nutrient values address a universal body given the age and gender, the food-based recommendations are made to reflect the national food stock and food culture, and, as such, the recommendations are already integrating different food ontologies [12].

The Finnish nutrition guidelines are authored by the National Nutrition Council, which has operated since 1954. The Council has a chairperson (alternating between the Ministry of Agriculture and Forestry and the Ministry of Social and Health Affairs), a vice-chair, 14 members, and a full-time general secretary, who is located at the Finnish Food Authority. The Council issuing the 2014 recommendations still included representatives from various stakeholder groups in a corporatist manner (more on this below), but now it consists solely of experts from ministries, independent research organizations, and health and nutrition authorities. Expert organizations nominate their candidates for the Council, which is appointed by the Ministry of Agriculture and Forestry for 3 years at a time. Interviewed council members mentioned how the placement of the Council under this ministry has been disputed over the years, with civil society actors claiming that the agricultural industry's influence has resulted in a productionist emphasis in the Council's work. Several of our interviewees declined this claim, however, one even pointing to the advantages of the administrative affiliation: the Ministry takes policies coming from its own organization more seriously. This was to say that while the Ministry of Agriculture and Forestry traditionally emphasizes domestic production over environmental concerns, it also wants to be involved in the sustainability debate.

The Finnish nutrition guidelines lean on the Nordic guidelines, published every eight years. Some members of the Finnish nutrition council have participated in the preparation of the Nordic Nutrition Recommendations, facilitating their adoption into the Finnish context. The National Nutrition Council adapts the Nordic guidelines to an individual level, to the national food culture, and to the specific public health issues faced in Finland.

The process of making the dietary guidelines has evolved over time, and is somewhat different for the general and the group-specific guidelines. As all the members of the National Nutrition Council have full-time employment elsewhere, much of the preparatory work for the general guidelines is carried out in ad-hoc expert groups at the Finnish Food Authority and the Finnish Institute for Health and Welfare. For the latest 2014 guidelines, an expert working group consisting mainly of nutrition and health experts was appointed to prepare the guidelines. The draft was then discussed, and finally accepted and issued by the National Nutrition Council. With regard to the group-specific guidelines, theme-specific experts are included in the process. For example, the guidelines for school meals were prepared in close collaboration between the National Nutrition Council, the Finnish National Agency for Education, and the National Institute for Health and Welfare. The different guidelines have also different legal basis and regulatory status. The organizing and provision of school catering are based

on legislation, several collective agreements, and the national core curricula for pre-primary education and basic education [13] (p. 17). As school meals are also considered a "multidisciplinary learning unit," their recommendations include elements beyond strictly nutrition.

When the theme of sustainable foods was introduced in the 2012 Nordic Nutrition Recommendations, some of the Council members thought sustainability belonged automatically to the Finnish guidelines as well, while others did not want to complicate the nutrition focus of the guidelines with other themes. The eventual inclusion of sustainability in the national guidelines resulted from several factors. One aspect was the topicality of climate issues on public agenda. As one interviewed nutrition scientist said, "[environment] has started to interest nutrition scientists, and they have started to collaborate with environmental scientists. And it all comes down to climate change." Moreover, the idea of eating seasonally had been discussed earlier among the Council members, and there was broad interest in including local and organic food in the guidelines, as they were in the interest of the then government. Some of the interviewees saw, however, a longer historical development in the scope of dietary guidelines: "It was from nutrition to food, and then also to physical activity. So, the environment had changed in a way that we do not look at things one by one but now we look at the whole, and it is clear that when we talk about food, there are also environmental issues involved."

Another interviewed Council member saw the Finnish membership in the European Union as a turning point for national food policy, and consequently to nutrition guidelines: "I think the new thinking on food policy started in '95 when Finland joined the EU and we needed to rethink our food system, as the national boundaries didn't hold anymore and there started to be import. [ . . . ]. A few years ago, the first meeting of nutrition council I attended, it was like when we talk about nutrition we talk strictly about the biochemistry that happens, what you need to stay healthy. But today no responsible expert can speak that narrowly, the recommendations have consequences and the consequences should be such that they promote societal wellbeing."

According to interviewed Council members, the inclusion of sustainability in the 2014 guidelines was not well received by the food industry and producer organization representatives. This is how an industry representative explained their position: "The industry values the nutrition recommendations, they are expected and emphasized in the product development. But concerning the inclusion of sustainability, the industry was a bit concerned, thinking that it will be confusing if there are too many perspectives. Is it going to be animal welfare issues next, and why not then for example the security of supply?" The discord within the Council was resolved when the chair of the Council made a reference to the program of the then government, which had sustainability as one of its spearheads. Even though the interviewees claimed that the National Nutrition Council works as an independent expert body, it was the reference to the government program that allowed it to include sustainability into the guidelines.

After the decision to include sustainability, the question was how precisely to write the sustainability perspective into guidelines with a health focus. An outside expert from the Natural Resources Institute Finland was invited in the writing process, since the members of the Council, being mainly nutrition experts, felt the topic was beyond their remit. According to interviewed Nutrition Council members, the aim was, at first, to integrate the sustainability perspective throughout the guidelines. This line was dropped as there were perceived to be many uncertainties regarding the different aspects of sustainability of food, and finally, a separate section on sustainable food choices [11] (pp. 40–43) plus an appendix were included.

The Nutrition Council issuing the 2014 recommendations was the last to include representatives from industry. The difficult discussions regarding sustainability contributed to the decision to make the Council an expert body, although the critique of its corporatist composition is of older origin. The broad representation had its pros and cons, as expressed by this Council member: "When the nutrition council is representative, the development of the nutrition perspective gets slowed down by the representatives' perspectives. But to be able to implement the recommendations, surely we need some kind of representation." One interviewee pointed to the changes in the nature of industry

representation as facilitating the change: "I think [industry representation] worked before [ ... ]. The field was more organized and through the Finnish Food and Drink Industries' Federation it was possible to reach the whole Finnish food industry. And the appointed representatives contributed with their expertise on, e.g., technology and markets, which the other members of the council did not have. But this changed around the time when the sustainability issue was discussed. The representatives were just making demands without having anything to contribute or without giving arguments. At the same time the industry field had got more fragmented, so they did not even represent the whole industry."

The decision to make the Council non-representative pertained also to broader developments in the Finnish food policy sector. An upper-level working group on food policy, the Food Policy Committee, was established by the Ministry of Agriculture and Forestry in 2016 to develop and coordinate food-related policies in central government. It consisted of representatives of different ministries; producer, advisory, and nature conservation organizations; and trade and industry [40]. The National Nutrition Council was placed under the Food Policy Committee as one of its three sub-committees. Although excluded from the Council, the industry could be satisfied as they got representatives in the upper-level Food Policy Committee. In addition, the Nutrition Council developed new forms of stakeholder collaboration to ensure the uptake of the recommendations: drafts for guidelines are published for comments from the industry and the public, and round tables and seminars are organized for stakeholders to bring in broad expertise on given topics. The interviewed council members and industry representatives were very positive on these new forms of collaboration. It seems that even though diversity within the Council was reduced, the change into an expert-based council allowed for increased ontological multiplicity. The scientist members of the multidisciplinary Council were comfortable with discussing different perspectives and epistemologies, and when the political demands of the industry representatives were externalized, it freed the discussion culture within the council. With the change, the council could claim impartial normative authority, being able to ascertain critics for not being 'on the leash' of food industry. What changed with leaving the industry representatives out of the Council was not the composition of policy actors, as the stakeholders were still involved in the consultations, but rather the process. In the new working model for the National Nutrition Council, an even broader range of views could be heard than before.

As the critique of the Council having a productionist emphasis highlights, the Council has historically been institutionally ambiguous. The introduction of sustainability highlighted the existing ambiguities and also brought new ambiguities, as the issue of sustainability broadens the food policy field. The recent reorganizing of food policy actors with the establishment of the Food Policy Committee seemed to bring clearer structure to the Finnish food policy sector, but institutional ambiguity still remains. The Food Policy Committee, which formally included the National Nutrition Council, was appointed on a project-basis until autumn 2019, and a new committee has not been appointed since. Also, some of our interviewees were skeptical whether the upper-level Policy Committee had any actual power, and maintained that the Nutrition Council remains the main food policy operative in Finland. It is unclear, however, whether it has a mandate to coordinate food policy more broadly, beyond nutrition, and there are different views whether food policy should instruct the dietary guidelines, or vice versa. According to an interviewee, "the recommendations should be a tool that is used in all nutrition and food policy." The government report on food policy from 2016 took sustainability as an overarching principle, but as the policy had several spearheads, with emphasis on domestic food security and export, it is not clear how sustainability fits with these various aims. The follow-up and implementation of the food policy are led by the Ministry of Agriculture and Forestry.

So far, nutrition and public health remain the focus of the National Nutrition Council, even though the importance of sustainability is acknowledged, as in the following statement by an interviewed nutrition scientist: "I think it's increasingly important to bring up these ecological issues and show people how to combine nutritional quality and ecological or environmental perspective. But the main focus of nutrition guidelines is in promoting public health, and the environment comes kind of after that." Even though the Council is now expert-based, for both the contents of the guidelines and

the process, it matters who the experts are, and the council seems to be also internally in a state of ambiguity. The Council members have ontologically differing views on how food should be considered in the guidelines: as nutrients, or as multifaceted components in the global food system. Consequently, the Nutrition Council becomes a site of ontological politics, deciding which version(s) of food is enacted in the guidelines and how the different ontonorms are emphasized. Expert-based policymaking may also narrow down and skew the ontological multiplicity involved in sustainable food policy and, for this reason, it is necessary to ensure sufficient stakeholder participation and coordination with other policies and broader political aims. The policy process should encourage dialogue and debate on the various political aims pertaining to food policy.

## 5. Ontonorms in Finnish Nutrition Guidelines

In this section we discuss the various ontonorms we identified in the Finnish dietary guidelines, our primary focus being on the population-level general guidelines from 2014. First, in Section 5.1, we briefly identify the ontonorms described by Annemarie Mol in her analysis of Dutch dietary advice, generally following the "Mind your plate" idea, but also introducing elements of the *Enjoy your food* approach [31]. We then move on to describe, in Section 5.2, how the concept of a sustainable diet was introduced in the 2014 recommendations, and the ontological multiplicity it brought to the dietary guidelines (Section 5.3). Finally, in Section 5.4, we discuss the two different versions of the sustainable ontonorm, the one we identified in the making in the Finnish guidelines, pertaining to the "Mind your plate" advice, and the other possible sustainable ontonorm, chiming in with a caring rather than cognitive-control approach. Table 2 summarizes the ontonorms, with examples from the documents, detailing the normative messages conveyed in the nutrition guidelines with their ontological commitments. The descriptions of the first three ontonorms are taken from Mol [25], the fourth from Vogel and Mol [31], and the two last ones are our own tentative descriptions of the sustainable ontonorms in the making.

### 5.1. The Conventional Dietary Ontonorms

The Finnish nutrition guidelines (listed in Table 1) rely on the same techniques and normative approaches as dietary counselling identified by Mol [25]. The first technique of dietary advice was counting in kilocalories the fuel provided by the food. This notion of food as energy has been a cornerstone of the Finnish guidelines as well. For example, in the 1987 guidelines, bodies were discussed in terms of energy balance, disturbances of which resulting in obesity. The relationship between food and body was formulated straightforwardly: "obesity is a direct consequence of an unbalanced diet."

The second technique consists of lists that distinguish between foods that are preferable, in the middle, or only to be consumed as an exception. The Finnish guidelines have since 1998 included a food pyramid, which divides foods into classes of preference. In 2012, the Nordic nutrition recommendations introduced a "traffic light" table of foods to be increased (green), replaced with more healthy options (yellow), or reduced (red) in the diet. This was also adopted in the 2014 Finnish guidelines. As in counting calories, dividing foods into good and bad implies that healthy food is not what people prefer if the body is let to consume what it desires. Health and pleasure are staged as being in tension, and, for instance, the 1987 recommendations state that "eating according to the recommendations requires information for making choices, and some amount of self-discipline."

The third technique in dieting advice Mol discusses is the disc of five, an illustration depicting five food categories that should be consumed regularly to receive the necessary nutrients. Finnish guidelines from 1987 and 1998 used the food circle as a model for a diverse diet. Since the 1998 guidelines, the disc has been further specified into a plate model, which is a depiction of the ideal composition of a single meal. The aim is to nudge habits in a healthier direction, increasing the consumption of vitamin-rich vegetables and decreasing the use of 'fast' carbohydrates and unhealthy fats.

**Table 2.** The 'conventional' (Mol 2014; Vogel & Mol 2014) and sustainability ontonorms mapped to content from the 2014 Finnish nutrition guidelines.

| Normative Message Concerning Eating | | Mind Your Plate | | Enjoy Your Food | Mind Your Plate and the Planet | Care for Yourself and the Planet |
|---|---|---|---|---|---|---|
| **Food/Body Ontology** | Biophysical | Epidemiological | Biochemical | Sensuous, Caring | Biophysical, Epidemiological, Biochemical, and Environmental | Ecological, Caring |
| | Food is fuel. Counting calories allows a rational mind to control a body seeking to eat more than it consumes. | Food is an input variable with health effects. Making lists of healthy and unhealthy foods helps to control a body desiring unhealthy foods. | Foods contain essential nutrients. Utilizing the plate model helps to ascertain a balanced diet. | Food is a source of pleasure. Pleasure is a crucial part of the body's feedback system. Bodies are capable of self-care when cultivated. | Adds *the planet* to the *Mind you plate* advice. Adds environmental flows to the biophysical, epidemiological, and biochemical models. Emphasizes cognitive control and rational choice. | Adds *the planet* to the *Enjoy your food* advice. From consumer to participant in food systems, e.g., in urban gardening. Emphasizes an ecological, caring approach towards the environment and self. |
| **Content in Finnish nutrition guidelines 2014 (examples)** | Losing weight requires limiting energy intake. Intake of energy nutrients specified. Energy density of a plant-based diet is low. | Recommended food choices, food pyramid. A traffic light –colored table of foods to be increased, replaced with more healthy options, or reduced in the diet. | Recommended intake of vitamins and minerals. The plate model. Mediterranean and Nordic diets as examples of balanced diet. | Healthy diet can be composed in many ways; the whole matters over single choices. Recommended diet is both healthy and tasty. Local food culture. | Converging health and environmental benefits of some foods listed. Excess consumption should be avoided; benefits of plant-based diet. Local food & organic food, although benefits are uncertain. | Not present in the 2014 guidelines. Examples in the recommendations for families (2016) and schools (2017), and the 'Food2030' policy document (2016): food sense, food joy. |

Although the Finnish nutrition guidelines are firmly following the "Mind your plate" route, the alternative "Enjoy your food" advice is not entirely absent. Moderation, versatility, balance, and enjoyability are recommended as starting points for a good diet in the older guidelines. In the 1987 edition, it is recognized that a balanced diet can be composed in many ways, and a diverse diet makes it possible for food to be both healthy and tasty: "A varied diet makes little sense if the food is not tasty. It is important to be able to enjoy the pleasure of eating without guilt and without suspecting the quality of food." The recommendations suggest that true enjoyment comes from eating healthy food, knowing it is good for the body. Curiously, in the most recent 2014 population-level recommendations, there is not a single explicit mention that food should be enjoyable.

Our findings from reading the Finnish nutrition guidelines show that they employ the same ontonorms Mol identified in Dutch dietary advice. Food is primarily discussed as fuel, as nutrients, and as good and bad for the body—the body in need of control. To eat right, people are perceived to require detailed nutrition information, instead of eating what they would prefer. Other meanings of food are noted, but the advice, in general, does not trust individuals to choose wisely, were they to follow the signals from their bodies. Only in the latest, group-specific food recommendations from 2017, which address children, are new openings re-adjusting the mainstream advice. We will return to them later.

### 5.2. Sustainability—Yet Another Motivation for Healthy Eating

While applying the same normative dietary advice as before, in including sustainability, the 2014 Finnish nutrition guidelines contain also a new kind of ontonormativity. Sustainability is presented in a short special section titled "Sustainable choices on the plate" [11] (pp. 40–43). The text discusses the sustainability effects of agriculture on a general level, and, more specifically, local and organic food, and it also looks at some key foods from a sustainability perspective. The main thrust of the discussion is that health and environmental benefits go hand-in-hand: shifting diets towards the recommendations would already as such reduce the environmental burden of food [11] (p. 53). For example, in terms of health, eating less meat reduces the risk of cancers and possibly type-2 diabetes and reduces the intake of saturated fat and excess energy. In terms of environment, it diminishes the carbon footprint and the eutrophication of lakes [11] (p. 54). The guidelines do not primarily instruct how to put together a sustainable diet, but present sustainability as an additional motivation for healthy eating. Another instance where environmental and health benefits of eating dovetail is when the guidelines mention the increased use of vegetables, root vegetables, potato, berries, fruits and cereals, and especially domestic seasonal foods as both sustainable and healthy. The guidelines have a long-standing interest to promote domestic production and national food items, and here sustainability is brought up as an additional argument for not only health but also other food policy goals (see next section).

A shift towards plant-based diets is broadly recognized as key in increasing the sustainability of food systems [6,41]. The various editions of the Finnish dietary guidelines have intermittently addressed vegetarian diets. In the 1987 edition, the vegetarian diet was not mentioned, but in the 1998 edition, the vegetarian diet was covered under "Special dietary recommendations." The text mentioned that "vegetarians avoid animal-based foods on health, ethical, or ecological reasons," but the connection between ethical and environmental issues and diet was not explored further. A successful shift from a mixed to a vegetarian diet was portrayed as not lightly recommendable, as it requires "strong motivation and enough enthusiasm, practical cooking skills, good knowledge of foodstuffs, and a change in taste."

In the 2014 general recommendations, vegetarian and vegan diets were discussed in a dedicated section and were no longer treated as a "special diet." The vegan diet is presented as a viable option, provided care is exercised with the intake of some micronutrients, but it is not endorsed as a preferable dietary option, despite environmental benefits. The text foregrounds the health benefits of the diet, stating that vegetarians are less overweight and have less cardiovascular diseases and type-2 diabetes than the Western population in general and that vegetarians also have lower blood pressure, lower total

cholesterol, and they live longer. In addition to diet, this is connected with vegetarians exercising more and smoking less than the meat-eating population [11] (p. 32). This focus on health bypasses other motivations to follow a plant-based diet, such as animal welfare issues or ecological grounds, and thus reduces other-directed ethical commitments to self-directed health interest. Not acknowledging other motivations than health behind food choices may undermine the authority of the dietary guidelines in the eyes of diverse publics.

Even though the guidelines aim at underlining the positive convergence of health and environmental aims, they also use a controlling discourse. An uneasy connection between excess eating, overweight, and sustainability is made: "Excess consumption should be avoided also for environmental reasons. [ . . . ] Since the energy consumption of overweight people is greater than with slender people and an increase in weight is always associated with excess energy intake, weight control measures are desirable also from the perspective of sustainable development." [11] (p. 12.) Maintaining normal weight is one of the main rationales of the nutrition guidelines, and countering 'the epidemic of obesity' is a key target of Finnish public health policy. Now environmental benefits are presented as a further motivation to avoid being overweight. Excess body weight is described both as a health and an environmental burden, and the quote suggests one's environmental performance can be directly read from the physical habitus, as a measure of the waistline. Not only are bodies perceived in need of discipline so that they do not eat themselves sick, but they are also ruled guilty in causing environmental problems. The message "Mind your planet is" added on top of "Mind your plate."

### 5.3. The Ontological Multiplicity of Sustainable Food

In her analysis of dietary counseling, Mol [25] explicitly bracketed off the globalized food system. This broader context is necessarily re-introduced with the discussion on sustainability in the guidelines. In the health paradigm, it is possible to think of the body as a closed system, with food, energy, and nutrients as input variables, and health or overweight-related illnesses as output variables. With sustainability, problems such as the eutrophication of lakes and climate change are brought to the table and onto the plates. It is no longer possible to ontologically define food as only energy or nutrients; sustainability brings in the whole food system, increasing the complexity and number of variables to be considered. Sustainability cannot be reduced to health and social sustainability; ecological sustainability is necessarily foregrounded when considering the sustainability of food production and consumption, which results in parallel and unresolved aims in the recommendations.

In general, a key message of Western dietary advice is "less meat, more veggies" [3,4,6,11,37]. This rule of thumb seems to join health and sustainability targets straightforwardly. In the national guidelines, however, a key tension is between individual-based health advice and broader aspects of sustainability that address the physical and social environment. The Finnish nutrition recommendations struggle with drawing the boundaries of the food system and with coordinating the ontological multiplicity of food. When considering the recommendation to reduce meat in national context, the issue inevitably draws in other policy fields and targets besides health and the environment. The 2014 dietary guidelines acknowledge the high environmental impact of beef production. The recommendation to reduce meat consumption is watered down in the text, however, as the environmental conditions in Finland are presented as suitable for cattle keeping. Environmental conditions are bundled with policy aims such as domestic production and maintaining livelihoods in the Finnish countryside. The continued rise in meat consumption is also deconstructed by pointing out that the increase has been mostly in poultry and pork, which have smaller environmental impacts compared to beef [11] (pp. 53–54). Here, national economic production interests are coupled with environmental relativism. Domestic production is portrayed as more sustainable than meat production elsewhere, and this reasoning does not consider other, ethical motivations to reduce meat consumption.

Under the section Sustainable choices on the plate, local and organic food are discussed in dedicated sections, as their promotion is in the interest of the Finnish government. The discussion adds to the ontological multiplicity of sustainable food. In discussing local food, the guidelines paint a vision

of bringing producers and consumers closer together, and of local production based on carbon-neutral, closed circles [11] (p. 42). In this scale, the familiar ontonorm of food as nutrients can be perhaps extended from the body to the local food system. Further on, the text mentions, however, that from the perspectives of neither health nor environmental sustainability are a place of significant production and transport. It does not become clear then whether local food is the preferred sustainable choice. Similar ambiguity pertains to organic food: while the guidelines describe how organic farming uses less chemicals and is better for biodiversity, it is also mentioned that organic products do not necessarily have health or environmental benefits, as the production may equally depend on fossil fuels as in conventional farming [11] (p. 43).

The dietary guidelines contain multiple ontological propositions on what sustainable food is. While in the health paradigm food is understood as energy and nutrients, sustainability does not limit itself to the question of what is sustainable food. The what-question is present in the overall message of eating less meat and more vegetables, which seems to neatly support both health and environmental sustainability. However, the question of sustainability is complicated by bringing in the questions of where (domestic/local production) and how (organic production) is the food produced. These questions connect the dietary recommendations to broader food policy questions and other policy sectors, with diverse and often conflicting targets. Health and sustainability are rivaled by the economic profitability of domestic production and the export of agricultural products. Furthermore, the recommended food choices aim at being culturally acceptable and economically accessible to the consumers, which complicates the where and how questions, as food production in Finnish latitudes is expensive. While environmental crises bring forth the importance of national self-sufficiency in food supply, the question of why remains largely unaddressed. According to a recent study, only a fraction of the global population can supply its demand for certain crops locally [42], and as such, sustainable food system cannot be based on closed nationalism; global supply chains and trade are needed for sustainable food systems, and it remains to be specified in food policies how this ties in with local solutions.

Environmentally sustainable food is enacted as ontologically multiple in the guidelines, and it sits somewhat awkwardly within the whole. The Finnish guidelines include components of a sustainable food system, but they are not organized into a coherent whole. Inclusion of sustainability brings new ontonormative openings to the dietary guidelines, suggesting different meanings of food and different ways to compose a healthy and environmentally sustainable diet, but as the guidelines fail to coordinate the multiplicity, they increase the normative burden of consumers to make responsible choices.

### 5.4. Mind Your Planet—Or Care for Your Planet?

While introducing sustainability, the Finnish dietary recommendations did not introduce any specific techniques to support sustainable eating. The norm seems clear—eat sustainably—but the ontological multiplicity of sustainable food is not unpacked into a clear choice architecture in the same way as with healthy eating. A more consistent approach would be to rearrange the relationship of health and environmental aspects of eating. This could comprise a version of the plate model based on the sustainability effects of various food items. Steps in this direction were taken in the first Finnish recommendations for school meals in 2017, issued by the National Nutrition Council, together with the Finnish National Agency for Education and the National Institute of Health and Welfare. The 2014 population-level guidelines discussed vegetarianism and veganism as special diets, but they did not represent a clear recommendation. The school meal recommendations go further in mainstreaming plant-based diets, providing a vegan plate model [13] (p. 37). Such supporting techniques of sustainable eating introduce the novel principle, "Mind your planet," which is not only about the effects of food in the body, but addresses also the effects of food choices on other people, other species, and the environment. The guidelines stress that a healthy and sustainable diet can be composed in a variety of ways, but the ontological multiplicity of sustainable food combined with

the logic of individual choice make the "Mind your planet" principle problematic, imposing more obligations on individuals.

The Finnish dietary guidelines include a further position, which chimes with Mol's notions of the enjoyment of food and care. This perspective is present in the most recent, group-specific food recommendations. Although the guidelines for families with children [13] do again focus mainly on the health aspects of food, they stress food education and allowing children to make choices on what and how much they eat, according to their age. The guidelines thus seem to fall between the regimes of choice and care—"Mind your plate" and "Enjoy your food"—in stressing information needed for composing a balanced diet, on the one hand, but also counting on the body being able to learn new tastes and healthier habits when properly cultivated, on the other. Furthermore, the guidelines for school lunches [14] do not only instruct in nutrition and meal composition, but school meals are considered part of the curriculum, supporting objectives related to education on food, health, manners, and a sustainable way of life. The school guidelines introduce new concepts into the dietary recommendation discourse, such as "food sense"—situational and experiential understanding of everyday eating and its social and cultural meanings—and "food joy"—combining tasty, nutritious, sustainable, healthy, and safe eating with positive food talk [14] (pp. 13–14). These techniques emphasize the sensory quality of food, such as textures, smell, and ambience, and stress the importance of the situated practices of eating. The approach follows the "Enjoy your food" advice [25], with the idea of cultivating the body to eat what is good for it. Ideally, sustainable food consumption would incorporate the many aspects of consumption, such as care, sociability, pleasure, social equality, and welfare [43].

Signals for new kind of sustainable food citizenship may be found also in the Government Report on food policy, "Food 2030," from 2016 [17]. In addition to issues related to food security and export, the report widely explored the environmental impacts of food production and consumption. Taking a systemic approach to food, it discussed sustainability, e.g., in connection to citizen participation and urban gardening. This integrated approach positions citizens differently from the traditional consumer approach, and provides the opportunity for food advice where bodies are not so much in need of control, but capable of participating in food production and care of the environment, learning new capabilities and sensitivities through, for instance, gardening. It is necessary to point out, however, that in emphasizing a "consumer is the king" message throughout, the policy report leaves food policy largely to the markets, and does not coordinate the ontological multiplicity of food any further than the dietary guidelines.

Dietary techniques leaning on guilt have proven not to be very efficient and may lead to counterproductive binge-eating [44]. There are differing views to what extent environmental guilt functions as a motivator for sustainable choices and action [45]. Analogically to dieting, an empowering approach to sustainable choices might work better than one based on guilt. It seems the Finnish nutrition advice is taking steps towards a caring approach to eating, especially when it comes to children. The next population-level dietary recommendations could link eating similarly to mental, physical, social, and ecological wellbeing, thus adopting a comprehensive sustainability approach. Instead of emphasizing the messages of "Mind your plate" and "Mind your planet," the guidelines could encourage to "Care for yourself and the planet," endorsing the ethical and social benefits of environmental sustainability and following lifestyles. Yet here again, it is good to keep in mind, as Vogel and Mol [31] also discuss, that there is a thin line between empowering people to enjoy all aspects of food and eating, and creating yet another norm to do so. While combined with the logic of choice, the ontological multiplicity of food puts the burden on the consumer, but in the logic of care the multiplicity, it works as a resource, allowing different modes of sustainable food practices and citizenship.

## 6. Discussion and Conclusions

We set out to analyze the role of sustainability in Finnish nutrition guidelines, focusing on the latest 2014 edition. In the analysis, we employed Annemarie Mol's concept of ontonorms, which

refers to the simultaneous enactment of ontological and normative commitments in various dietary techniques—different understandings of food, bodies, and proper eating. The analytical lens of ontonorms allowed us to tease out the ontological politics in sustainable dietary guidance and led us to pay attention to the emergent state and institutional ambiguity of Finnish food policy.

The Finnish 2014 population-level dietary guidelines seem undecided whether environmental sustainability is a societal concern to be overarchingly integrated into the guidelines, or a matter of individual preference. While the nutrition core of the recommendations has remained virtually unchanged, the food recommendations have diversified. The key message is that following the established techniques of healthy eating and conforming to the dietary norms outlined in the recommendations is also sustainable. No new techniques are presented for the quest of making diets more sustainable, such as a sustainable plate model or food pyramid. Still, sustainable eating is presented as a conscious consumer choice, and the guidelines fail to connect it to pleasurable and caring motivations that are important in mainstreaming environmental action and facilitating changes in eating. The "Mind your plate" advice is extended to "Mind the planet," but it is within the same parameters of individual choice and responsibility. However, the most recent Finnish guidelines for school meals have adopted elements that depart from the established approach. Notions such as "food joy" and "food sense" represent ideas that correspond to the "Enjoy your food" idea, and combined with sustainability could produce the advice to "Care for the planet."

The enacted ontological multiplicity of sustainable food in the Finnish dietary guidelines reflects the institutional ambiguity of food policy. Food policy is an emerging policy field, where many traditional policy sectors have a stake. Parallel objectives of what is sustainable food and where and how it is produced are present in dietary recommendations, which fail to prioritize and coordinate the different aims. While the lack of clear guidance may be confusing to consumers aiming to eat sustainably, the ontological multiplicity of sustainable food also opens up space for new politics with different actors and practices. In Finland, the National Nutrition Council is a key actor with regard to sustainable diets, but it is not clear whether the Council has a mandate to sustainability spearhead the dietary guidelines. Even when aiming at being an independent expert organization, the Council cannot escape becoming a site for ontological politics—a site where sustainable food is negotiated. The introduction of sustainability into the Finnish dietary guidelines can be considered a safe opening: the recommendations work as a testbed for sustainable food policy, being a non-binding policy instrument. But for the same reason, the recommendations cannot be the only instrument for sustainable food policy; a range of policy tools are needed for sustainable food transformation. Currently there is no political consensus on what is a sustainable diet, and neither is there a coordinated preference on the targets of sustainable food policy. Public debate is often polarized between domestic production targets and ecological concerns. In this situation, the expert-based Nutrition Council is forced to the role of policymaker, even though it prefers to see itself as implementing scientific knowledge. The issue of sustainable diets is thoroughly political, including fundamental questions of democracy, such as food justice. In this light, the central role of the expert Council in policy making is problematic, but at the same time its recommendations can be seen as important input to take the political debate on sustainable diets forward.

Unpacking the ontological multiplicity of a sustainable diet and the ontonorms of dietary recommendations sheds light to the politics and uncertainties involved. Plant-based diets are increasingly emphasized to reduce the environmental cost of food [46], yet only modest reductions to meat consumption or semi-vegetarian diets are recommended. Since 2014, however, there has been an unprecedented growth in interest towards plant-based diets [47]. There are also indications of growing acceptability of strong sustainable consumption governance [48]. These developments suggest that guidelines foregrounding sustainability, with a clear choice architecture, could be widely accepted. Strongly integrating sustainability would signal the recommendations being responsive to cultural and societal changes. What status sustainable eating will get in the next Finnish dietary guidelines depends importantly on the treatment sustainability will have in the next Nordic Nutrition Guidelines,

due in 2022. There are indications that the sustainability of food systems will have a central role in the forthcoming Nordic recommendations [16]. With the increasing awareness of environmental impacts of different foods and global calls for food transformation, sustainability could spearhead the dietary recommendations in Finland and elsewhere.

**Author Contributions:** Conceptualization, M.S. and M.J.; methodology, M.S. and M.J.; investigation, M.S.; data curation, M.S.; writing—original draft preparation, M.S.; writing—review and editing, M.S. and M.J. All authors have read and agreed to the published version of the manuscript.

**Funding:** This research was funded by the Academy of Finland, grant number 296702. M.S. was funded by Turku Institute of Advanced Studies and Kone Foundation during writing.

**Acknowledgments:** The authors wish to thank colleagues in the POPRASUS project "Politics, practices and the transformative potential of sustainable diets" for comments on earlier versions of the paper. We are also grateful to the anonymous reviewers for their helpful feedback.

**Conflicts of Interest:** The authors declare no conflict of interest. The funders had no role in the design of the study; in the collection, analyses, or interpretation of data; in the writing of the manuscript, or in the decision to publish the results.

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
