# Peer review of "Institutional Ambiguity and Ontological Politics in Integrating Sustainability into Finnish Dietary Guidelines"

_sustainability, doi:10.3390/su12135330_

Round 1
Reviewer 1 Report
This paper discusses Finnish nutritional guidelines and the ontological layers present in these guidelines with a focus on sustainability. The analysis is based on (i) a historical analysis of the guidelines through interviews with involved civil servants, experts and stakeholders and (ii) a text-based analysis of the guidelines themselves.
In doing so, this paper will make a relevant contribution to the field of food policy science. In depth analysis of case-based food policies is not yet a staple in this body of literature. The lens of using ontonorms is interesting and adds to the field. However, I do feel that the article can benefit from some revisions, with a focus on elaborating the implications of the presence of multiplicity of ontonorms for food policy. This will enhance the author’s stated objective of ‘Our theoretical contribution is to apply the perspective of ontological politics and ontonorms in policy analysis.’ My recommendation is to concentrate the effort of revising the paper on the following 6 main points.
Main points
- Section 1 and 2. Please state the research question explicitly. Lines 1767-171 capture the contribution, but not the research question. Articulating the research question might clarify what the exact contribution of the conclusions is. Please also discuss why the paper is relevant to people outside of Finland. What are generalizable implications for policy scientist and policy makers?
- At several moments in the paper, it is discussed that the national nutritional council used to include stakeholders, but this changed and now only includes experts. As the objective of the paper is to apply ontonorms to policy analysis, I think that elaborating on the change to experts-only and the implications for food policy and its ontology is warranted.
Regel 171 ‘need to be coordinated’ : What does this mean in the context of policy science? What extra insights can you provide, building on Mol’s work? What does coordination mean in the context of food policy?
Line 228 – 231: What is the normative and epistemological implication of this change, from stake holder participation to experts only? Did you see evidence that the ontological multiplicity was reduced?
Line 345-349 : “expert-based” : It matters a lot which experts are included. Reflect critically on this in relation to ontonorms. It could explain internal ambiguity, as you write in the following lines. This brings up the question whether ambiguity is a negative state or a positive state for policy? I think Mol would argue that it is neither, and part of the world as we experience it. Is there a need to seek ‘coordination’? Why so? Please reflect on the policy implications of expert-only policy making. See comment on line 171.
Line 596-610 please connect these thoughts to the discussion on the ‘expert-only’ model currently employed in Finnish food policy. What are implications for policy makers and politicians? Please develop the implications with greater depth. What is the role of other stakeholders in ontological multiplicity and the need for coordinating this ambiguity? - One of the conclusions drawn in the paper, is that institutional ambiguity is currently present in Finnish food policy. I feel that the authors need to clarify what exactly they mean in using this concept and what its implications are for the (effectiveness of) food policy.
Line 73: I don’t understand the reference to Hajer’s institutional void here, as the National Committee is hierarchically above the Nutritional Council, and both these bodies are part of the Ministry of Agriculture. I do understand from your writing that at present it is uncertain if the Committee will be reinstalled or that institutional organization of food policy within the ministry will revert to how things were before – but this not an institutional void in the sense that the policy problem transcends policy entities hierarchically on the same level and/or with distinct polities and therefor "no one is responsible". In the Finnish case, it seems clear to me that ultimate responsibility for food policy lies with the Minister of Agriculture. Maybe the concept of institutional void is not needed to draw the conclusions of the paper.
Section 4, starting line 217. The conclusion (‘institutional ambiguity’) seems to apply at the present. How would you describe the institutional arrangement in the past? Since you draw on the history of becoming the guidelines. Is the current ambiguity consistent with how things were in the past (pre-2014) or is it something new? Is it really because of introducing the topic of sustainability into the nutritional guidelines? Or are the other possible causes, that could be traced in past guidelines (before 2014). - Improve and clarify section 3 (Materials and methods).
Table 1 does not convey the centrality of the 2014 document. Instead, it is confusing that the table is not organized chronologically. I think the table would be improved if the hierarchy of importance of the documents in the research is shown.
Line 187/188 and 205/206 please clarify how the ‘word processing’ was done. Include a reference to qualitative text analyzing methods. If a computer program was used (e.g. Atlas.ti), please clarify.
Line 209-210: was there any external check of the ontonorms you found? Cross reading between authors? E.g. check with interviewees. At least recommendation for further research. I would strongly encourage to do a review of results by cross checking the findings between authors. - Please improve the organization of section 5. When I read section 3, I expected a list or table with identified ontonorms in section 5. Perhaps a table of ontonorms can be provided, including references to documents and prime examples. Moreover, I feel that section 5 could then be organized better to structure it according to the ontonorms identified. In the current write-up, I felt a bit lost as to where the information was going and what key results (i.e. the identified ontonorms) were.
Line 351- 390. Please clarify which document in which year is referenced. I found this paragraph not easy to read, alternating between formal references and describing by title/year. The latter would have my preference. Refer to Table 1 as well. - Please check English grammar in this section, especially the (translated) quotes. See ‘minor comments’.
Minor comments
Section 1 and 2. Please include references to similar research that identifies ontological multiplicity in nutritional guidelines that are using a slightly different angle. E.g. Korthals (2017), Ethics of Dietary Guidelines: Nutrients, Processes and Meals, J Agric Environ Ethics 30:413–421
Line 126 -131 You need to give an argument why Mol’s approach is flexible enough. Now it reads as just your opinion.
Line 248: correct is ‘with regard to ..’ (as regards)
Line 388 ‘bodies’ do you mean ‘people’ or ‘individuals’?
Line 417-421 It was unclear what point you want to convey. The role of vegetarianism? Or other themes that were included in the guidelines? Why are these points relevant to sustainability (the caption of this section)? Maybe it is just a matter of explaining with a few more words.
Line 461. A key message of dietary advice globally is “less meat, more veggies”. Please provide a reference supporting the statement. I don’t think the statement is true, as there are many areas in the world where the advice would be to increase protein consumption (e.g. Subsaharan Africa). The EAT Lancet diet, which can be viewed as a global nutritional advice, does discern different recommendations for different parts of the world. However, one of the main critiques on the EAT Lancet diet is that is foregrounds a Western / developed world perspective with a ‘less meat, more veggies message’, thereby side-staging the rest of the world. On second reading, I suspect that maybe ‘in general’ instead of ‘globally’ was meant. In that case, please do take my comment to check English language and grammar seriously
Line 456-457 Please substantiate this claim. Does it follow from the text references discussed in paragraph 5.2?
Line 470-475 I think these lines provide an example of the general claim in line 456-457. An explicit connection would improve the section.
Reviewer 2 Report
I found interesting to read the article. However, there are few issues and corrections that can be addressed to improve the article.
Line 91-96: Too long sentence, reorganize the sentence
Line 102-103: Not clear, revision needed
Line 128-130: Not clear, revision needed
Line 144: follow a biochemical approach, need examples
Line 192: several groups
Line 197: Table 1- 2014- better to translate into English in parentheses for the international readers
Line 219-220: Not clear, revision needed
Line 233: Citation?
Line 235: Who conducted the interviews? How? no details
Line 318: Citation?
Line 329: Council was
Line 362-363: Sentence-confusing, revision needed
Overall comments:
Better to describe initially why this article is important (significance and outcome)
Discuss the methods properly, the interview and data analysis process
